# Functionalization of Silicone Surface with Drugs and Polymers for Regulation of Capsular Contracture

**DOI:** 10.3390/polym13162731

**Published:** 2021-08-15

**Authors:** Omar Faruq, Pham Ngoc Chien, Nilsu Dönmez, Sun-Young Nam, Chan-Yeong Heo

**Affiliations:** 1Department of Plastic and Reconstructive Surgery, Seoul National University Bundang Hospital, Seongnam 13620, Korea; ofaruq1991@gmail.com (O.F.); ngocchien1781@gmail.com (P.N.C.); nlsdonmez@gmail.com (N.D.); 2Department of Plastic and Reconstructive Surgery, Seoul National University Hospital, Seoul University College of Medicine, Seoul 03080, Korea

**Keywords:** silicone, surface modification, polymer, drug, capsule contracture

## Abstract

Breast reconstruction is achieved using silicone implants, which are currently associated with major complications. Several strategies have been considered to overcome the existing limitations as well as to improve their performance. Recently, surface modification has proved to be an effective clinical approach to prevent bacterial adhesion, reduce capsular thickness, prevent foreign body reactions, and reduce other implant-associated problems. This review article summarizes the ongoing strategies for the surface modification of silicone implants in breast reconstruction applications. The article mostly discusses two broad categories of surface modification: drug-mediated and polymer-based. Different kinds of drugs have been applied with silicone that are associated with breast reconstruction. Initially, this article discusses studies related to drugs immobilized on silicone implants, focusing on drug-loading methods and their effects on capsule contracture. Moreover, the pharmacological action of drugs on fibroblast cells is considered in this section. Next, the polymeric modification of the silicone surface is introduced, and we discuss its role in reducing capsule thickness at the cellular and biological levels. The polymeric modification techniques, their chemistry, and their physical properties are described in detail. Notably, polymer activities on macrophages and inflammation are also briefly discussed. Each of the reviewed articles is summarized, highlighting their discussion of capsular thickness, foreign body reactions, and bacterial attachment. The aim of this review is to provide the main points of some research articles regarding the surface modification of silicon, which can lead to a decrease in capsular thickness and provides better patient compliance.

## 1. Introduction

Silicone is a common material and has various uses in sealants, lubricants, medicine, cooking utensils, thermal insulation and surgical application [1,2]. Recently, in plastic surgery, silicone has been widely used for breast reconstruction and breast augmentation [3]. Using silicone in plastic surgery has various benefits, such as its low toxicity and anti-adhesive properties. However, its use in this field has been associated with capsular contracture, which occurs when the collagen fiber capsule shrinks, tightens, and compresses, which is the result of a prolonged inflammatory response after the insertion of silicone into the human body. This results in pain, discomfort, and the unsatisfactory appearance of the breast. Capsular contracture is still recognized as one of the most common and serious complications of breast augmentation and breast reconstruction [4]. Foreign body reactions to silicone, hematoma, peri-implant infection, and bacterial contamination as a result of breast implants are considered the main factors associated with capsular contracture [5].

In the history of silicone breast implants, continuous efforts have been made to reduce complications, especially capsular contracture. Clinical evidence has indicated the importance of modifying the design of silicone in order to improve its performance after implantation. Several modifications were applied, but they ultimately failed to overcome the limitations. In 2012, the FDA approved silicone gel implants made by Silimed, which are known as “gummy bears” [6]. The aim of using this gel was to provide protection against leakage during breast implant rupture. However, safety concerns have arisen regarding the composition of the gel [7]. Other silicone implants produced by Inamed, Mentor, and Silimed were approved by the FDA. However, Inamed implants developed by PIP were taken off the market due to early rupture. Their silicone grade was not suitable for the human body [8,9]. Currently, the most prominent companies in the breast implant market are Groupe Sebbin, Mentor Worldwide, LLC Allergan, GC Aesthetics, Sientra Inc., Laboratoires Arion, Hansbiomed Co. Ltd., Groupe Sebbin SAS, CEREPLAS, Silimed. AllERGEN (Textured and Smooth), MENTOR (Textured and Smooth), and IDEAL implant (Smooth).

The global breast implant market is gradually expanding and is expected to earn a revenue of USD 4.9 billion during the period of 2019–2026 [10]. The market is divided based on shape, product application, and region. Concerns regarding the prevalence of breast cancer and capsular contracture have an effect on the breast implant market. Moreover, the post-surgical implication and the high cost significantly hinder the growth of the market. Currently, in the breast implant market, the round shape is preferred over the anatomical shape. The round-shaped breast implant’s market size is USD 1845.2 million and is expected to significantly increase. Round-shaped implants are available in a wide range of sizes and proportions to breast shape. However, the anatomical breast implant market is also growing as a result of the need to balance breastfeeding as well as issues related to breast volume and asymmetry. The breast implant industry also uses implants for breast reconstruction and augmentation. The market for breast augmentation has shown rapid growth, as, compared with the procedure for breast implantation, breast augmentation is less complicated and requires a shorter period of time [11]. The surface characteristics of silicone implants, such as a smooth texture, are also gaining further attention.

Silicone breast implants mainly consist of two parts: filling materials (silicone gels) and the shell (silicone elastomer). The gel acts as filler materials where their viscosities depend on the polymer chain and crosslinking. High-molecular-weight silicone is biologically inert, making a filler gel approximately 1000 centistokes [12]. Moreover, for early curing of low-molecular-weight silicone, gel content that has a high chance of diffusion from the implant is incorporated. The diffused gel contains low-molecular-weight siloxane ranging from 3 to 20 (molecular weight: 200–1500) [13]. On the other hand, the texture roughness of the shell can vary. According to the ISO, the surface roughness of smooth, microtexture, and macrotexture implants is below 10 µm, between 10 and 50 µm and above 50 µm respectively [14].

Smooth-surface silicones were the earliest type of silicone implant. These silicones continue to be used now for breast implants, but some previous studies demonstrated the high rate of formation of capsular contracture [15,16]. Later, textured-surface silicone implants were developed to improve fibrosis inhibition. However, these biomaterials led to other problems, such as the formation of double capsules and late seromas in breast augmentation surgery [17]. Next-generation implants, namely microtextured-surface and micronanotextured implants, were developed with a surface roughness of 10–100 µm [18]. These commercial silicone implants have exhibited no double capsules or late seromas, with a low rate of reoperation as well as a lower rate of capsular contracture in an in vivo study [19]. However, these silicone implants have only recently been developed: the clinical data are still not sufficient to support their use, and the remaining limitations of surface implants remain to be overcome.

The prevention of capsular contracture formation is the most important challenge in plastic surgery. Recently, some studies have investigated the reduction of the formation of bacteria on surface implants. Based on the properties and the continued limitations associated with silicone, there have been various efforts to design optimal implant surfaces. The modification of the implant surface with antibacterial coatings, antimicrobial coatings, and prophylactic antibiotics, and the modification of the physical properties of the material, were successful in reducing the risk of bacterial contamination [20,21,22,23]. In addition, various methods were utilized to reduce capsular contractures, such as modification of the semi-permanent surface and the use of microtextured structures that regulate the interaction between the surface of silicone and the immune system of the host. Consequently, the degree of early capsular contracture in the breast can be reduced [24,25]. Moreover, surface modification also diminished the acute inflammation resulting from foreign body reactions to silicone implants. However, the level of capsular contracture is still different depending on the surface implant [26,27].

In this review, we discuss several recently developed methods of functional modification of silicone implants with drugs and polymers to prevent fibrosis after implantation into the body.

## 2. The Procedure of Fibrosis Formation

The implantation of biomaterials into the body causes an inflammatory response followed by the wound healing response. However, the phenomenon of fibrosis formation takes place via an immune reaction comprising six steps. The contribution of cellular activity in each step depends on the duration of the immune reaction. The formation of fibrosis is the result of the release of soluble factors of macrophages in the immune cells where the fibroblasts are stimulated. The procedure involves injury, blood–biomaterial interactions, provisional matrix formation, acute inflammation, chronic inflammation, the development of granulation tissue, foreign body reactions, and fibrosis capsule development [28].

Blood–biomaterial interactions occur when implants are inserted and come into contact with the leakage of blood at the wound site. This interaction results in protein absorption to the surface of biomaterials and causes adverse reactions such as the activation of coagulation and leukocytes that produce inflammation, adhesion, and the activation of platelets [29,30].

Provisional matrix formation is considered the result of the deposition of blood proteins on the biomaterial’s surface. Then, the provisional matrix forms (Figure 1).

Acute inflammation often occurs in a short time, usually lasting less than one week. It depends on the level of injury, the organ where the biomaterial is implanted, and the level of formation of the provisional matrix. Various inflammatory cells contribute to this inflammatory stage. Neutrophils (known as polymorphonuclear neutrophils, PMNs) secrete an array of proinflammatory cytokines, such as interleukin (IL)-1α, IL-1β, IL-6, IL-7, IL-18, and tumor necrosis factor (TNF)-α [32]. Mast cell degranulation is observed to contribute to inflammatory mediators (i.e., histamine, tryptase, leukotrienes, serotonin, heparin, endothelin, nitric oxide) as well as the release of IL-4 and IL-13 [33,34]. These inflammatory cytokines are used to determine the subsequent progression of the foreign body reaction after implantation (Figure 2).

Chronic inflammation is the result of persistent inflammation with the presence of mononuclear cells such as monocytes, lymphocytes, and macrophages as well as the proliferation of blood and connective tissue formation at the implant site [35]. Chronic inflammation can last for weeks to months or years and this problem depends on the level of injury. Many factors can cause chronic inflammation, such as the chemical and physical properties of implant biomaterial. The mononuclear cells secrete IL-8, IL-18, IL-4, IL-7, IL-10, and other cytokines (Figure 3) [36,37]. Fibrosis is caused by chronic inflammation or is the result of failed wound healing.

The foreign body reaction occurs during the chronic inflammatory step when the implantation of a foreign biomaterial into the body leads to the development of inflammation and fibrotic processes, where the foreign body giant cells (FBGC) appear at the surface of the biomaterial. During a foreign body reaction to the biomaterial, various cytokines are secreted, such as IL-1, IL-6, IL-8, TNF-α, macrophage inflammatory proteins (MIP)-1, MIP-2, MIP-3, and IL-13 [38,39,40]. The expression of these factors contributes to the formation of capsules as well as modulating fibrosis severity.

The development of fibrosis capsules is the last phase of the foreign body reaction and healing process, and fibrosis is formed. In this stage, the synthesis of collagen, ECM remodeling, and the reduction of the stimulation of the biomaterial reaction were observed, associated with various cell types, including immune cells, fibroblasts, endothelial cells, keratinocytes, and adipocytes (Figure 4) [41]. However, fibrosis can lead to capsular contracture and scarring when the implantation of the foreign biomaterial into the body is not stabilized, leading to persistent infection and tissue injury.

## 3. Surface Modification Using Drugs

Surface modification techniques are applied in various fields. In the medical field, although various types of surface modification have been developed, they still pose some challenges for researchers. Recently, some studies have focused on the modification of biomaterial surfaces to overcome limitations that prevent capsular contracture formation. In this review, we introduce some currently used methods and the results of silicone implant modification with drugs under study.

### 3.1. Montelukast

Montelukast is known as an inhibitor, approved and first used in clinical practice 20 years ago [42], that inhibits CysLTs production in the inflammatory phase. Montelukast is considered a safe drug with no harmful effects.

Leukotrienes (LTC), known specifically as LTC4, LTD4, and LTE4, are related to the inflammatory cascade and have an important role to play in the development of capsular contracture [43]. When silicone is implanted in the body, the acute inflammatory phase is initiated by the infiltration of PMNs from the blood vessels at the site of implantation [44]. Subsequently, the PMNs produce cysteinyl leukotrienes (CysLTs), which are a family of potent inflammatory lipid mediators, synthesized through the breakdown of arachidonic acid of eosinophils and a variety of inflammatory cells such as mast cells, dendritic cells, smooth muscle cells, monocytes, and macrophages [45,46]. Chronic inflammation can occur due to the constant presence of the silicone implant, and CysLTs stimulate the migration and proliferation of fibroblasts (Figure 5).

Among recent research, the surface modification of silicone implants via drug delivery was studied and achieved some expected results. The main purpose of this technique is to maintain sustained drug delivery and avoid an immune response. Some studies of the surface modification of biomaterials with drug loading were carried out in various medical fields. In particular, in plastic surgery, Kim et al. showed, in their research, four different types of silicone implant, namely intact silicone implants, silicone implants coated with PLGA only, silicone implants coated with montelukast, and silicone implants coated with both PLGA and montelukast [47]. The silicone implants were sprayed twice with different conditions of drugs, with an interval time of 30 min. The studies were performed in vivo in a rat model. The implants with montelukast demonstrated significantly decreased production of polymorphonuclear leukocytes, which are related to the secretion of CysLTs. The silicone implants coated with montelukast and the silicone implants coated with both PLGA and montelukast sustained the release of montelukast for 3 and 15 days. Additionally, the number of fibroblasts also significantly decreased. This led to the inhibition of TFG-β expression, and the amount of myofibroblasts could be decreased by the inhibition of CysLT production. Moreover, the thickness of the capsule and the intensity of collagen were significantly reduced. These results were observed in a previous study [48]. This demonstrates that the use of the surface modification of implants with montelukast is effective at reducing the development of breast capsular contracture.

### 3.2. Tranilast

Capsular fibrosis is the result of acute and chronic inflammation when an interaction between silicone implants and the body occurs. The initiation of capsular fibrosis occurs through several steps, and the last step is fibrosis. Acute inflammation lasts for a few days. It depends on the size, shape, and physicochemical properties of the silicone implants [5]. During chronic inflammation, various cytokines are secreted by macrophages. Specifically, TGF-β is known to mainly contribute to the procedure of fibrosis during chronic inflammation, and TGF-β is observed during platelet degranulation at the acute inflammation phase (Figure 6) [49,50,51]. This contributes to the formation of a thick fibrous capsule around the silicone implant.

Tranilast is an antiallergic drug. It was developed and launched in 1982. Tranilast is used in the treatment of inflammatory diseases including allergic rhinitis, asthma, keloid scars, hypertrophic scars, and allergic pink eye [53]. Tranilast is known as a drug that inhibits the release of TGF-β from keloid fibroblasts, as well as the release of inflammatory cytokines such as TGF-β1, IL-1β from monocytes and macrophages [54]. However, tranilast was also shown to inhibit the synthesis of collagen by fibroblasts and to inhibit TGF-β-induced extracellular matrix synthesis [55,56,57,58].

Based on the effect of tranilast, preventing the formation of capsular contracture around silicone implants may be possible through the inhibition of the activity of TGF-β. Recently, some studies were conducted using coating with a biocompatible polymer, poly(lactic-co-glycolic acid) (PLGA), and tranilast for the prevention of capsular contracture. The silicone implant was separately coated on all surfaces twice, with an interval of 30 min [59], and the silicone implant was coated with dots containing PLGA and tranilast [60]. Interestingly, the results showed that the combination of coated dots on silicone implants was more stable than loading drugs on all surfaces. In combination with PLGA in each coating dot, the tranilast could be continuously released for more than 14 days. The silicone implants coated with a mixture of PLGA and tranilast displayed a decrease in capsular thickness, and the density of collagen and the expression of TGF-β also were observed to be reduced. These results were similar between the two methods of surface modification with drugs. These current approaches for the surface modification of silicone implants with drug delivery could be considered for further studies to prevent capsular contracture.

### 3.3. Triamcinolone

Triamcinolone is a glucocorticoid, approved in 1958 by the FDA, that is used to treat a variety of diseases, such as skin diseases, allergies, and rheumatic disorders, and it is particularly known for its use for preventing capsular contracture resulting from inflammation. Triamcinolone is considered to be an anti-inflammatory drug. Triamcinolone can inhibit the formation of inflammatory cytokines IL-1 and IL-6 in humans [61], prevent the activation of TNF-α release from cells of monocytes and macrophages [62] and block the transcription and translation of IL-1ß, IL-8, and TNF-α at the gene level (Figure 7) [63].

Capsular contracture is still considered one of the most common and serious complications of the implantation of silicone when chronic inflammation occurs. Various drugs have been developed to control capsular contracture. However, the long-term use of drugs can lead to adverse effects. As a result, specific applications and drug delivery by loading the drugs onto the surface of biomaterials such as silicone are essential in plastic surgery and other fields. Recently, various studies involving effective methods for drug delivery on the surfaces of biomaterials were carried out to increase its activity and stability for desired applications such as surface modification with enzymes, peptides nanoparticles, and antibacterial drugs [65,66,67]. These studies uncovered a diverse range of effects.

In the field of plastic surgery, silicone implants often have a large size, compared to other fields. Beom et al. [68] investigated the loading of different concentrations of triamcinolone acetonide on the surface of silicone to control capsular contracture. Different concentrations of triamcinolone acetonide were sprayed on the outer surface of the shell silicone implants (Figure 8). The spraying procedure was performed two times with an interval of 30 min (Figure 8). In the in vivo rat model, the silicone implant loaded with triamcinolone acetonide showed sustained release of the drug for 12 weeks, and the release rate could be controlled by the amount of loaded drug in the silicone implant. In addition, in the drug-loaded silicone implant, the thickness of the capsule and the intensity of collagen were observed to decrease, compared with a non-coated silicone implant. These results showed that the inflammation and the expression of proinflammatory cytokines were inhibited by the effect of triamcinolone acetonide. Because of this, the number of fibroblasts and myofibroblasts, strongly linked to pathogenic fibrosis, also was reduced. With control of the released amount of triamcinolone acetonide on the silicone implant, this could be a promising method for preventing capsular contracture.

### 3.4. Itaconic Acid

Itaconic acid is a well-known organic compound containing unsaturated dicarboxylic acid and has been associated with the immune system. At an industrial scale, it has gained much interest for polymer synthesis as it involves a double bond of its methylene group. It is generally obtained from the fungus *Aspergillus terreus*; however, further research is ongoing to increase the production of itaconic acid for mass application [69]. Recently, itaconic acid’s effect on immune-responsive gene 1 protein (IRG1) was studied in mammalian cells. The study showed that the IRG1 protein was upregulated in macrophages under proinflammatory conditions; however, the complete role of itaconic acid during inflammation was not revealed. Another crucial feature of itaconic acid is its antimicrobial activity, which contributes to an efficient immune response together with anti-inflammatory metabolites and cytokines. It inhibits the glyoxylate shunt of invading pathogens, which is essential for the survival of pathogens [70]. Moreover, itaconic acid inhibits bacterial growth by the inhibition of two enzymes, methylisocitrate lyase (MCL) and isocitrate lyase (ICL). In addition, itaconic acid blocks propionyl-CoA carboxylase, leading to reduced acetic acid and propionic acid assimilation in bacteria (Figure 9).

Silicone breast implants carry a high risk of bacterial contamination and fibrous capsule formation around the implant. It was demonstrated that an excessive foreign body reaction (FBR) caused by a silicone implant might result in CC formation. Bacteria on the breast implant influence the formation of a biofilm by interacting with the adjacent environment, which eventually increases the thickness of the CC. Surface modification of silicone implants with itaconic acid is an innovative approach for reducing bacterial contamination and foreign body reactions. PDMS surfaces coated with IA showed better antibacterial activities by reducing bacterial and protein adsorption (Figure 10) [71]. IA-conjugated PDM S (IA-PDMS) and IA–gelatin-conjugated PDMS (IA-GTpoly-PDMS) were developed and evaluated for their antimicrobial as well as anti-inflammatory activities. The protein absorption results showed significantly lower values in the presence of IA, which was crucial to inhibit biofilm formation. Moreover, IA-containing PDMS decreased the inflammation around the implant for up to 4 weeks. Notably, IA-conjugated PDMS showed higher-density collagen compared to IA-GTpoly-PDMS for up to 8 weeks. The researchers further evaluated the fibroblast and myofibroblast presence in the implant site by immunostaining. The results showed a small number of myofibroblasts present around IA-GTpoly-PDMS, with no reduction in fibroblasts.

### 3.5. Halofuginone

Halofuginone is an anti-fibrotic drug frequently applied with silicone implants for overcoming capsule formation. This drug is a collagen-I synthesis inhibitor with TGF-β signaling interferences [72,73]. It can covalently bond to silicone implants by simple dip-coating (Figure 11) [74]. The authors found no systemic side effects and observed reduced CD 68^+^ and TGF-β, collagen type I and type II, and capsular thickness after 3 months. The deposition of collagen can be reduced by an inhibitory intervention in the TGF-b cytokine signaling pathway.

## 4. Surface Modification Using Polymers

In tissue engineering, polymers have been widely applied as coating materials and to tailor surfaces to be more biocompatible, less cytotoxic, as well as biodegradable. Polymers’ physical and chemical properties are favorable for the modification of implants. Recently, polymers have been widely applied for bioactive molecule delivery in a sustained manner for longer periods. In addition, shape memory polymers with tunable mechanical action have gained interest for surface coating.

Polymer coatings on silicone are rapidly increasing due to several advantages. In terms of breast implants, it is necessary to provide suitable biointerfaces, maintaining their properties and controlling integration with host tissues. These behaviors can be achieved using appropriate polymer coatings on the silicone implants. Recently, researchers have applied myriad polymers on silicone’s surface based on their biocompatibility, flexibility, and mechanical properties. From nanoparticles to hybrid coatings, polymers can be applied in various ways to ensure the development of a strong attachment to silicone. Polymeric modification of silicone plays a vital role in reducing inflammatory cytokines, antibacterial attachment, and capsule contracture.

In the following section, the latest developments in the modification of silicone implants with polymer coatings are briefly discussed, highlighting their role in capsule contracture formation. Moreover, the coating methods and the properties of modified silicone are considered in this section. The effects of polymer coatings on different kinds of cells, inflammatory cytokines, and tissue repair sites are described in detail, with an emphasis on polymer coatings that function to reduce CC.

### 4.1. Natural Polymers

#### 4.1.1. Spider Silk

Spider silk was applied on silicone implants for reducing capsule formation. Silk is generally an excellent biocompatible material with appropriate mechanical properties [75]. More importantly, the silk protein is safe, without dose-limiting toxicity or immune reaction. The simple dipping method was used for silk coating and the result showed a coating thickness of approximately 900 nm [76]. After coating, 1M kH_2_PO_4_ solution was applied for the fixation of the coating, which resulted in the formation of a B-sheet for the water-insoluble silk coating. Further, the study showed that the proliferation of human fibroblasts was significantly reduced on the silk-coated implant. Unlike silicone with fiber networks, flat-like silicone showed less protein absorption and was less attractive for fibroblasts. Most interestingly, the human monocytes were attached both in silk-coated and uncoated silicone implants; however, their differentiation into CD68-positive macrophages regarding the silk-coated silicone was significantly declined. The silk-coated implants significantly reduced the capsule thickness, post-operative inflammation, and contracture formation. The coating effectively masked the surface implant during the first few months of implantation; in addition, the expression of follistatin, basic fibroblast growth factor (bFGF), and connective tissue growth factor were reduced, indicating reduced fibrosis (Figure 12).

#### 4.1.2. Interleukin-4(IL-4)

Capsular contracture is strongly related to inflammation at the implant site. Long-term inflammation may lead to fibrous tissue formation and subsequently result in capsular contracture. The control of a host/immune reaction, as well as chronic inflammation, is desirable. IL-4 is a cytokine that inherently reduces inflammation by the control of macrophages activities. IL-4 immobilization on the silicone implant is an effective way to reduce foreign body reactions and fibrous tissue formation. The effect of IL-4 on macrophage polarization during silicone implantation was studied, and its function in fibrous capsular formation was evaluated [77]. The results showed that IL-4 significantly promoted macrophage polarization in vitro and in vivo. It was demonstrated that IL-4 reduced the expression of inflammatory cytokines IL-6 and TNF and increased anti-inflammatory cytokine (IL-6 and Arg-1) expression. Furthermore, animal studies showed that macrophage polarization may help to reduced capsular thickness, tissue inflammation, and myofibroblasts infiltration.

### 4.2. Neutral Hydrophilic Polymers

#### 4.2.1. Poly(glycerol monomethacrylate)

Copolymers composed of two poly(glycerol monomethacrylate) (PGMMA) terminal blocks and a central poly(dimethylsiloxane) (PDMS) block were studied on silicone implants. The deposition created smooth and stable surfaces, which dramatically modified the protein absorption behavior of silicone substrates. The coated polymers in a water environment affected protein absorption, transforming the surface characteristics from preferentially fibrinogen-absorbing to preferentially albumin-absorbing. This switch can be very beneficial for decreasing cell adhesion and activities [78].

#### 4.2.2. Polyethylene Glycol

Silicone is suitable for application in various biological fields due to its low costs, ease of fabrication, and mechanical properties. However, a high degree of hydrophobicity frequently leads to the aim of the application not being fulfilled. Hydrophilic coating on silicone is a measurable technique to overcome extreme hydrophobicity. Dong et al. showed that the coating of PEG on silicone implants significantly reduced the adsorption of proteins on the surface of the silicone [79]. In this study, three types of protein were used to evaluate the adsorption properties, namely, bovine serum albumin (BSA), lysozyme (Lys), and Rb IgG/FITC.

### 4.3. Zwitterionic Polymer Coating

#### 4.3.1. Methacryloxyethyl Phosphorylcholine (MPC)

Sunah Kang et al. developed silicone implants that could reduce the fibrous capsule with a dense coating of 2-methacryloxyethyl phosphorylcholine (MPC) polymer. MPC is a zwitterionic polymer that mimics the head group of phosphatylcholine lipids in the plasma membrane and is widely applied as a coating material on orthopedic, cardiovascular, and ophthalmologic medical devices [80,81]. Heat-induced polymerization was applied for covalently grafted MPC around the silicone implant. Compared to UV-induced polymerization, heat-induced polymerization produced thicker MPC layers (Figure 13) [82]. The water angle results showed that the MPC polymer-modified the silicone surface from hydrophobic to hydrophilic. Moreover, protein adsorption, as well as fibroblast studies, demonstrated the performance of coated silicone implants compared to non-coated ones. The MPC-grafted implant showed adsorption on silicone surfaces that was reduced by 55% and 64%, respectively. NIH-3T3 cell adhesion studies revealed that the MPC-grafted implant had fewer cytotoxic effects. Since fibroblasts activated by inflammatory cytokines are responsible for capsular formation around the implant, the lower adhesion of fibroblasts on MPC-grafted silicone decreased the capsule thickness. The implantation of silicone implants in a pig model showed that the MPC-grafted silicone implant had lower capsule thickness at 8 weeks and 24 weeks. Furthermore, the anti-fibrous effect of the MPC-grafted implant was demonstrated by inflammation-related protein expression [83].

#### 4.3.2. Poly(carboxybetaine methacrylate) and Poly(sulfobetaine methacrylate) (pSBMA)

Zwitterionic polymers carboxybetaine and sulfobetaine are receiving a considerable amount of attention as antifouling materials. These polymers contain both positive and negative groups in a repeated unit. Moreover, they have unique properties, as they possess a neutral charge containing a water molecule. Strong bonding to water on their surface drastically reduces protein adsorption as well as that of other molecules. Coating of PDMS surfaces with zwitterionic polymers, poly (sulfobetaine methacrylate) (pSBMA), and poly (carboxybetaine methacrylate) (pCBMA) reduces biomolecule adsorption and foreign body reactions [84]. In this study, photografting/photocrosslinking was used to graft the polymers and evaluate their antifouling properties. Notably, nonspecific protein adsorption and fibroblast adhesion were significantly reduced in the coated surface compared to the uncoated one. The results of these studies revealed that the modulation of the fibrotic response was crucial for reducing the occurrence of capsular contracture.

## 5. Surface Modification Techniques

### 5.1. Layer-by-Layer Deposition Techniques (LBL)

Complications associated with breast implants are mostly capsule contracture and induced cancer at the implant site. Recent studies have shown that a smooth surface is favorable for capsule formation, whereas a textured surface reduces the capsule thickness. However, a textured surface is primarily responsible for cancer formation [85]. One of the important strategies to resolve this problem is a dual modification of silicone implant, where one layer is modified physically and the other layer is modified chemically (Figure 14).

In a study, the first microtextured layer was prepared by uniform-sized microparticles on the implant before curing. The size of the micropattern of the textured implant was controlled by the particle size. Notably, the micropattern size was smaller than 100 μm (Figure 15). The micropattern with a size ranging from 70 to 100 showed significantly reduced SMA expression as well as fibroblast activities and capsular formation. However, the textured surface was related to BIA-ALCL; polymeric modification of the smooth surface was achieved by layer-by-layer deposition of poly-L-lysine and hyaluronic acid. Both the nontoxic polymers showed synergistic application after silicone implantation. Studies showed the suppression of TGF-cytokine release from fibroblast cells against coated silicone. On the other hand, the fibroblast marker vimentin and myofibroblast marker SMA were evaluated in terms of their activities in a dual-coated implant. Unlike the uncoated silicone implant, the coating layer dramatically changed the distribution and morphology of cells, without aggregation on the implant. Moreover, lower expression of SMA in regard to the coating demonstrated that the differentiation of fibroblasts to myofibroblasts was significantly suppressed. The dual-layer polymer coating reduced capsule thickness; however, compared to the textured-surface polymer, the coating did not reduce the capsule thickness significantly. Both polymers, PLL and HA, are natural polymers that easily degrade, showing lesser effects in vivo [87,88]. The LBL coating showed a synergetic effect on fibroblast counts, myofibroblast counts, and collagen density [89]. The synergetic application is promising for the modification of silicone implants [86].

### 5.2. Microgrooved Pattern

The microgrooved pattern is another technique for the surface modification of silicone implants, improving biocompatibility and reducing capsule formation. Carbon-ion implantation with microgrooved silicone surface did not alter the cell adhesion, while cells were arranged in a more orderly manner, which subsequently reduced the capsule formation. Modification of silicone resulted in a hydrophobic surface, which improved cells’ adhesion and distribution. Moreover, in a long-term period (30 days), the microgrooved surface pattern showed a lower number of inflammatory cells and less collagen around the implants, delaying capsule contracture formation. Although these studies showed that a microgroove-patterned silicone surface with C-ions reduces the incidence of capsule contracture, the underlying mechanism was not elucidated [90].

### 5.3. Dot Pattern

The stability of polymers and their sustained release pattern is necessary for reducing capsule contracture. During surgery, usually, the implant should be folded and crumpled for insertion; thus, a coating on the entire surface would be subjected to severe mechanical stress, with a high chance of breakage or loss [91,92]. Dot pattern strategies showed that a stable polymer coating on the silicone surface leads to the local, sustained released of tranilast. In vitro drug release studies demonstrated an initial burst due to the high distribution of the drug near the polymer. Afterward, the drug was released in a sustained manner for 14 days. The result revealed the benefit of using a PLGA matrix coated with dots, which released the drug in a diffusion-mediated manner [59].

### 5.4. Drug Delivery Net (DDN) Method

Drug delivery with silicone implants is always challenging. A polymeric network for sustained release and diffusion from the implant still has limitations. Moreover, clinically, silicone implants are made in different sizes; thus, a specific coating process is currently needed. An elastic net plays a significant role in optimizing the size-specific coating on silicone of different sizes. Polyurethane has excellent mechanical properties for the synthesis of a drug delivery net, where it provides suitable and sustained released of triamcinolone [93]. Since polyurethane is mechanically stretchable, the net was tightly wrapped around the silicone implant. The significant benefit of net wrapping was that it could cover samples ranging from small to large sizes. The result of the studies revealed that DDN showed drug release in a sustained manner for 4 weeks. The release of the drug was similar between the intact strained conditions. The studies evaluated the drug effect for the same DDN system in both large and small silicone samples and demonstrated that both types of the sample showed a similar effect regardless of the sample size.

## 6. Conclusions

Silicone implants require significant improvements to enhance their action, and multidisciplinary research has been undertaken to investigate their physical and chemical surface modification. New strategies and modification techniques are drastically enhancing the development of advanced silicone implants. Our study highlighted the literature related to the surface modification of silicone implants with drugs and polymer coatings. We summarized the different modification techniques together with their roles in the cellular and biological environment. Various drugs and polymers are being modified to overcome the major complications that occur after silicone implantation. This review article discussed in detail the drug and polymeric modification of the silicone surface and the efforts to find a solution to reduce capsule contracture and inflammation. However, a suitable option for surface modification remains elusive; comprehensive future research seeks to overcome these major problems.

## Figures and Tables

**Figure 1 polymers-13-02731-f001:**
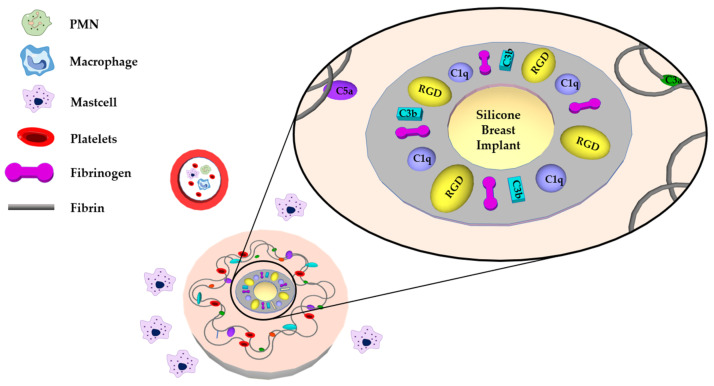
Schematic illustration of Phase 1 after implantation-initiated foreign body reaction and synthesis of a matrix around silicone breast implant. Reproduced with permission from [31].

**Figure 2 polymers-13-02731-f002:**
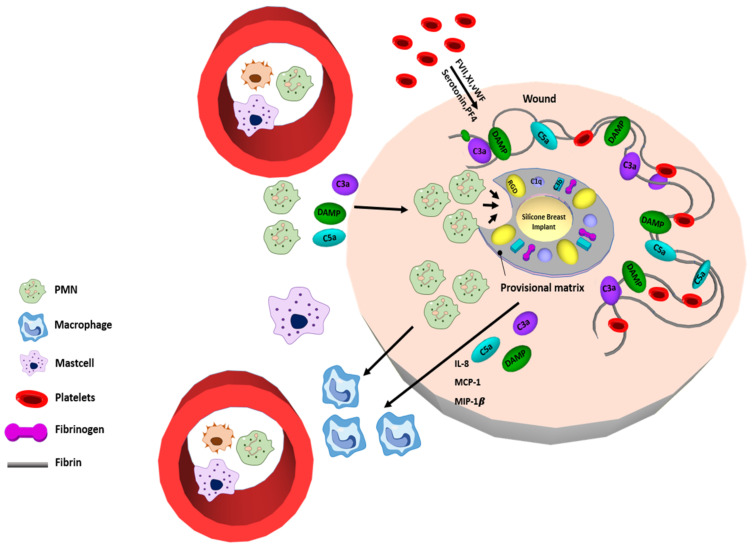
Schematic illustration of Phase 2 after implantation. Acute inflammatory stage with availability of polymorph nuclear leukocytes (PMN, neutrophils), mast cell and presence of macrophages at the implantation site. Coagulation factor and fibrin infiltrated at the implant site. Reproduced with permission from [31].

**Figure 3 polymers-13-02731-f003:**
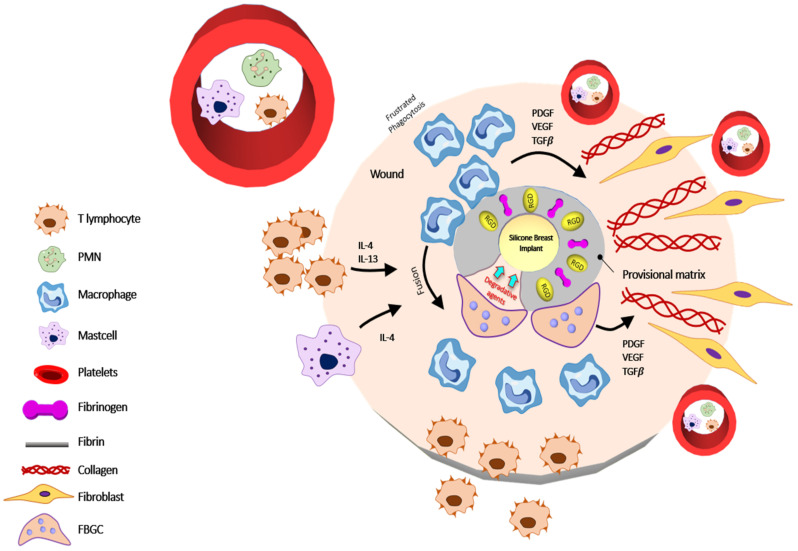
Schematic illustration of Phase 3 after implantation. Chronic inflammation begins with the dominance of macrophages and lymphocytes, which secreted the cytokines IL-4, IL-13. A longer period of chronic inflammation and impaired wound healing leads to the development of granular tissue formation. Reproduced with permission from [31].

**Figure 4 polymers-13-02731-f004:**
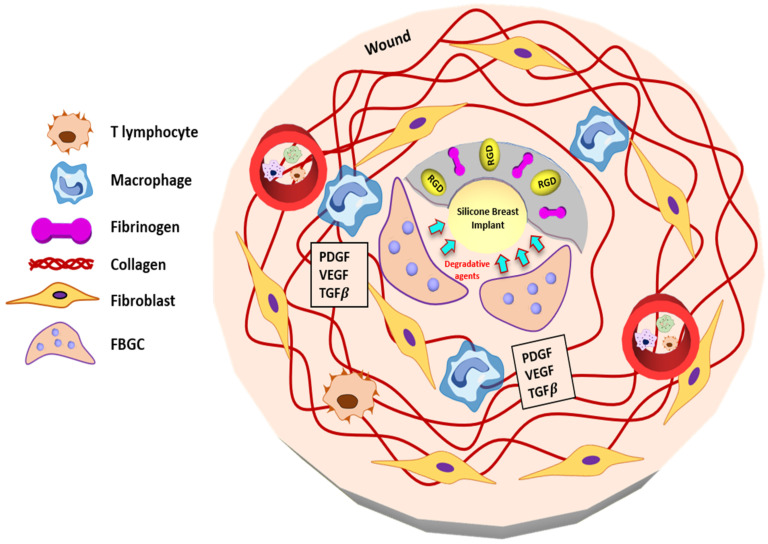
Schematic illustration of Phase 4 after implantation. Foreign body reaction eventually leads to the formation of the fibrous capsule. Various cytokines secreted were responsible for fibrous capsule density as well as severity. Synthesis of collagen, ECM synthesis around the silicone implants at the end of this stage. Reproduced with permission from [31].

**Figure 5 polymers-13-02731-f005:**
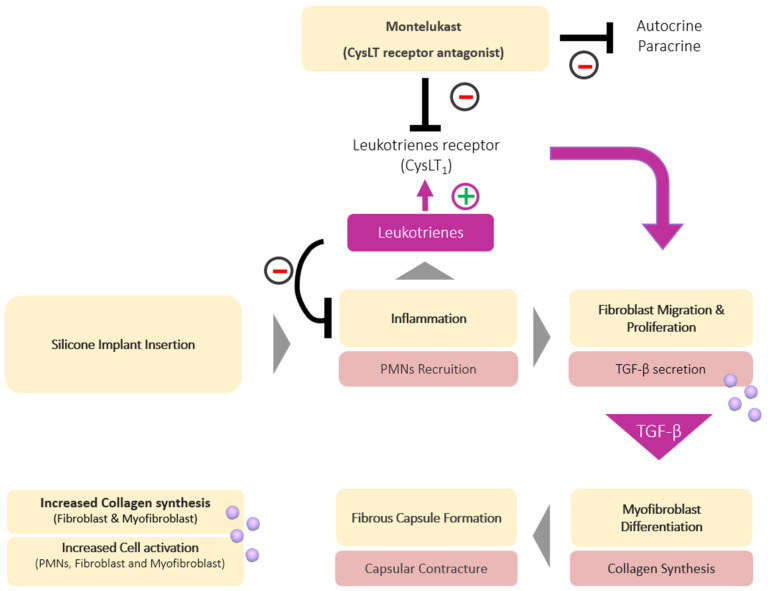
A representation of mechanism through which montelukast reduces the capsule contracture related to silicone breast implant. It acts as a CysLTs inhibitor which reduces chronic inflammation subsequently decreasing the fibrous capsule formation.

**Figure 6 polymers-13-02731-f006:**
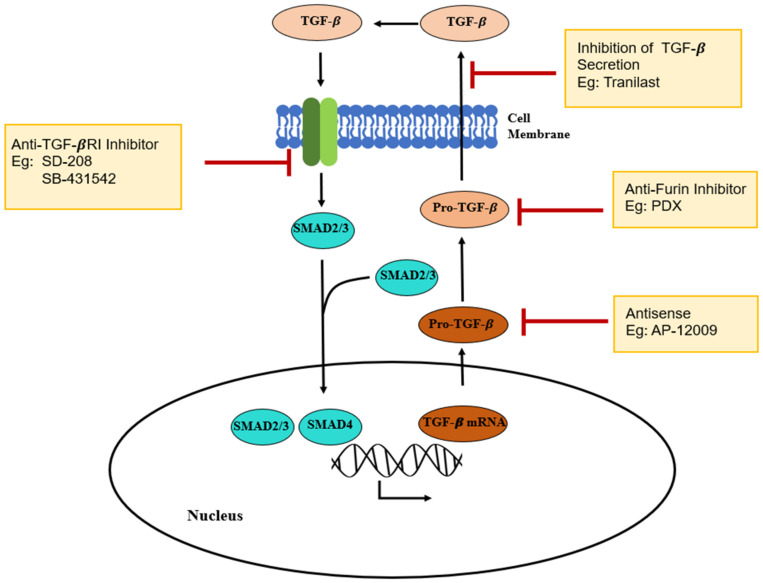
Molecular mechanism of tranilast action. TGF-β is responsible for prolonged inflammation and causes the fibrous capsule formation. Tranilast plays a role as an inhibitor of TGF-β. Reproduced with permission from [52].

**Figure 7 polymers-13-02731-f007:**
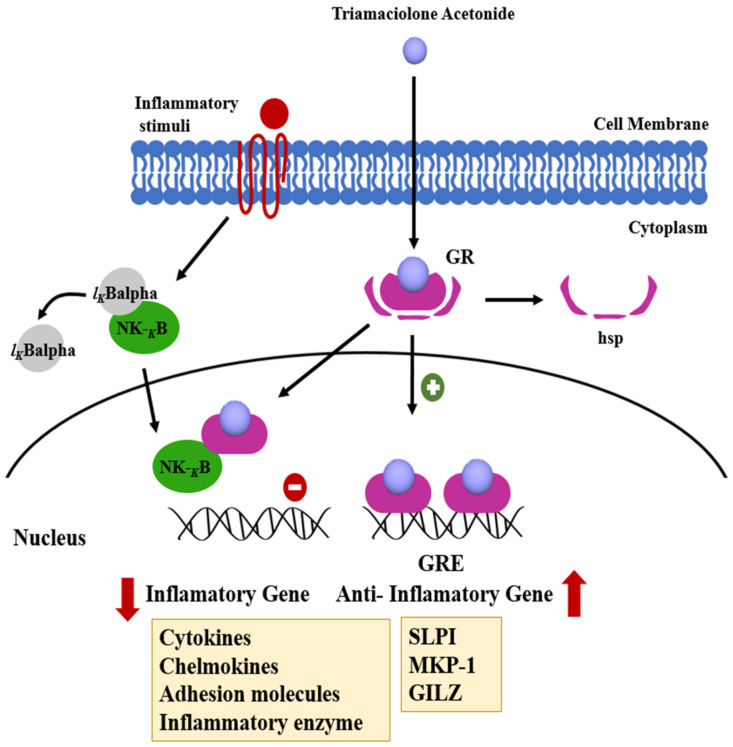
Molecular mechanism of triamcinolone acetonide action. It can reduce the expression of cytokines responsible for capsule formation. Reproduced with permission from [64].

**Figure 8 polymers-13-02731-f008:**
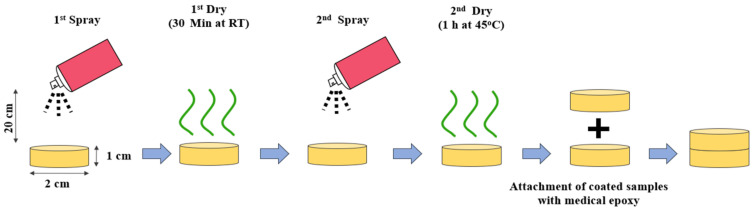
Spray drug coating technique on the silicone surface. Initially, drug solution was applied on the silicone surface followed by dry 30 min at RT. After drying, the drug solution was sprayed a second time and then dried for removal of the residual solution [68].

**Figure 9 polymers-13-02731-f009:**
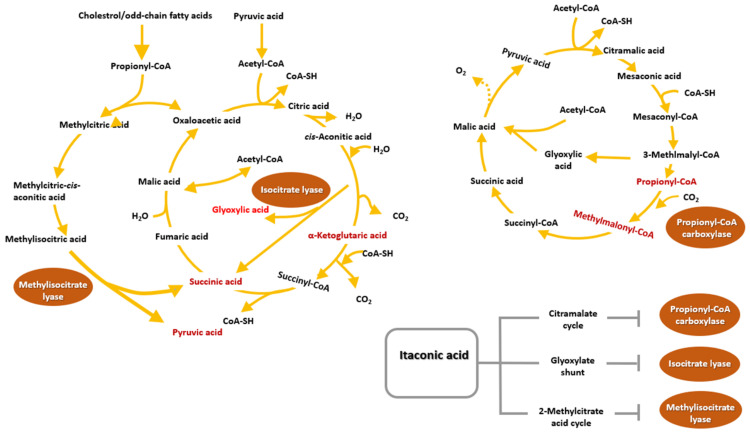
The role of itaconic acid on anti-microbial activities. It inhibits key enzyme isocitrate lyase that has methylisocitrate lyase activity in Mycobacterium tuberculosis. It also inhibits the 2-methyl citrate cycle. The citramalate cycle was halted by inhibition of propionyl-CoA carboxylase in the proteobacterium Rhodospirillum rubrum. Enzymes inhibited by itaconic acid are enclosed in a circle. Reproduced with permission from [70].

**Figure 10 polymers-13-02731-f010:**
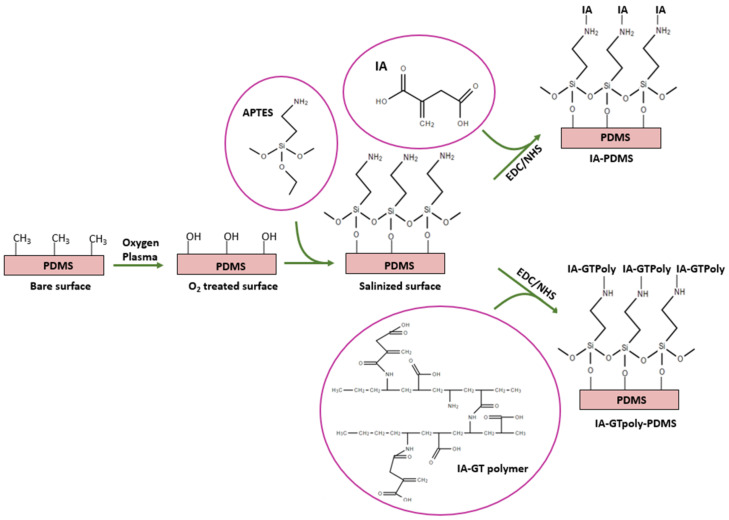
Surface modification of silicone implants with itaconic acids (IA). Gelatin was used to conjugate the IA on the PDMS surface. IA and IA-GT polymer both immobilized on the PDMS surface and evaluated their performance in terms of sustain delivery [71].

**Figure 11 polymers-13-02731-f011:**
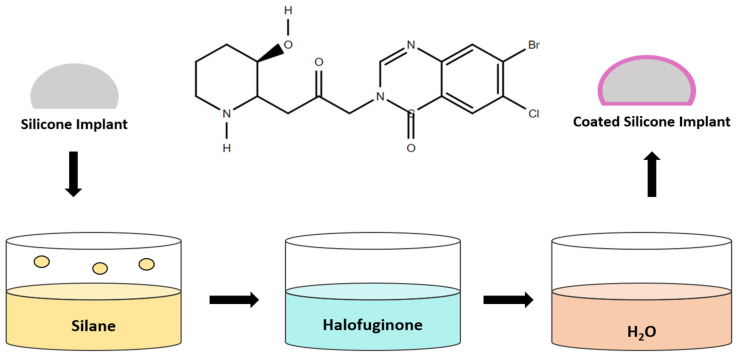
Dipping method for modification of silicone implants with Halofuginone. This is a simple method with a cost-effective approach. Halofuginone modified silicone implant significantly reduced the collagen I synthesis, which is crucial for fibrous capsule formation. Reproduced with permission from [74].

**Figure 12 polymers-13-02731-f012:**
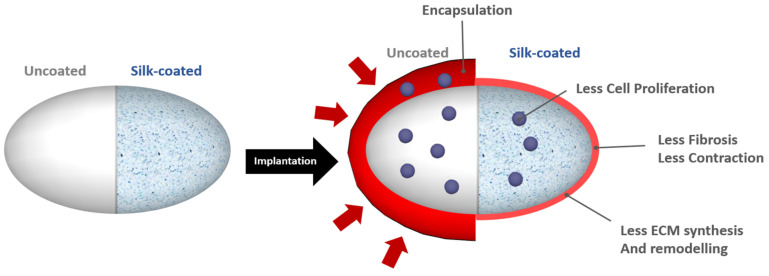
The schematic representation of silk coating on silicone implant and their performance on capsule contracture formation. Silk reducing the cell proliferation and ECM synthesis around the breast implants. Reproduced with permission from [76].

**Figure 13 polymers-13-02731-f013:**
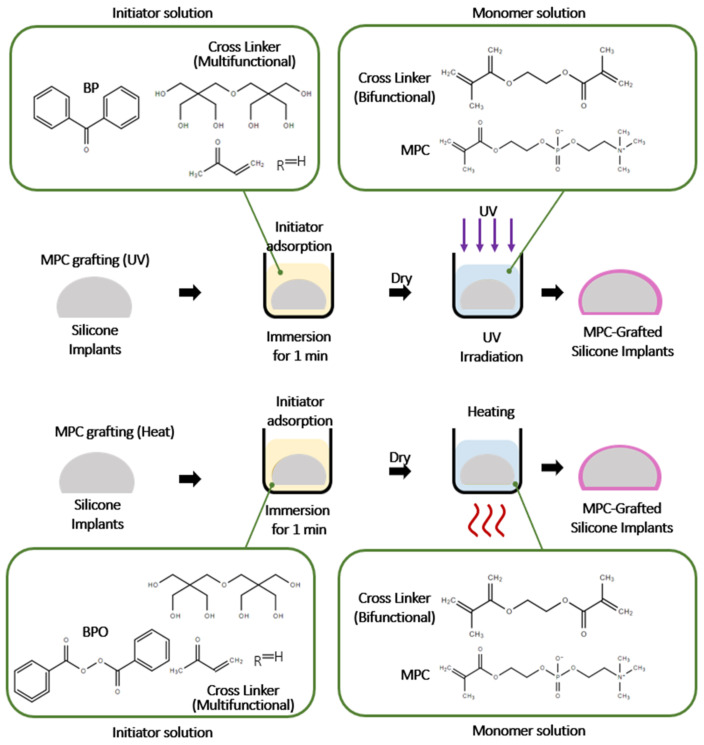
Surface grafting method of silicone implant with methacryloxyethyl phosphorylcholine (MPC). The polymerization grafting on the implants was conducted with 15 min-UV irradiation and 16 h heating at 70 °C. Reproduced with permission from [83].

**Figure 14 polymers-13-02731-f014:**
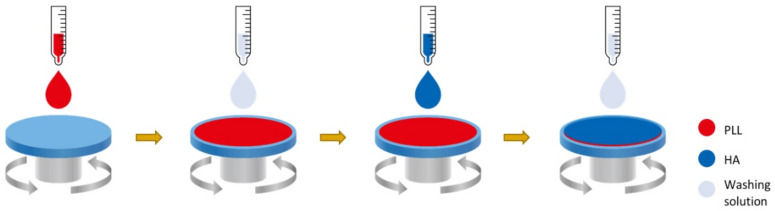
Preparation method of a dual polymer layer on the silicone implant with Hyaluronic acid (HA) and poly-L-lysine (PLL). The first and second coating layers consist of PLL and HA, respectively. Both the coating layers were uniformly distributed on the silicone implant and exhibited synergetic effects on capsular fibrosis formation [86].

**Figure 15 polymers-13-02731-f015:**
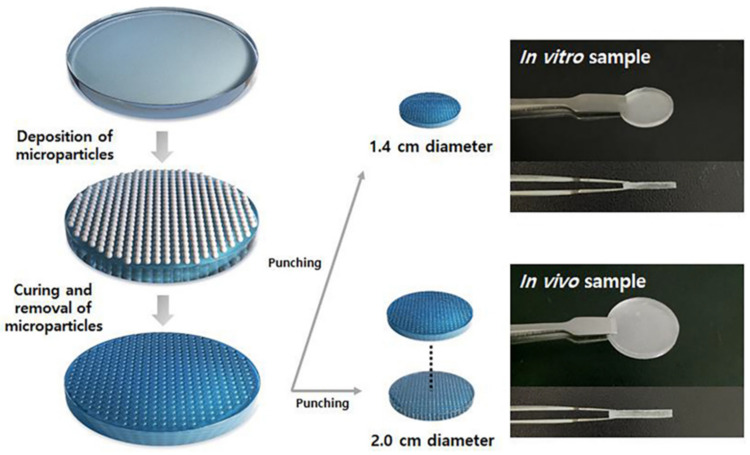
Preparation of the concave hemisphere pattern on the PDMS surface [86].

## Data Availability

Not applicable.

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
