# Peer review of "Functionalization of Silicone Surface with Drugs and Polymers for Regulation of Capsular Contracture"

_polymers, 2021, doi:10.3390/polym13162731_

Round 1

Reviewer 1 Report

It is a comprehensive review about functionalization of silicone surface with drug and polymers for regulating capsule contracture by Faruq and Chien, and coworkers. I recommend it for publication in Polymers after the following points are addressed.

  1. Format issue, the title ‘Functionalization of Silicone surface with drug and Polymers for regulating capsule contracture.’ should be changed to ‘Functionalization of silicone surface with drug and polymers for regulating capsule contracture’.

Line 229-230, why the sentence is yellow highlighted?

Line 313, ‘Polymers’ should be changed to ‘polymers’.

  1. Line 62-64, different kinds of surface modification of silicones have been reported. Several studies (doi.org/10.1016/j.actbio.2011.02.001; doi.org/10.1021/la303438t; doi.org/10.3390/polym11020305) are recommended to be included regarding to this point.
  2. The resolution of figure 4, 6, and 13 should be improved.
  3. There is no proper figure caption for figure 4-6.
  4. Proper copyright statement should be added in some of the figure caption.
  5. Figure 9, some of the text is covered by the arrow.
  6. Section 4 is the essential section in this review since Polymers is polymer journal. The authors should improve this section to a higher level. For example, this section is divided into spider silk, poly(glucerol monoethacrylate), and methacryloxyethyl phosphorylcholine (MPC), which is not proper. I suggest it could be divided into three categories: natural polymer (proteins, polysaccharide, and so on), neutral hydrophilic polymer (PEG, poly(glucerol monoethacrylate, and so on), and zwitterionic polymer (poly(methacryloxyethyl phosphorylcholine), poly(carboxybetaine methacrylate), and so on).

Author Response

Format issue, the title ‘Functionalization of Silicone surface with drug and Polymers for regulating capsule contracture.’ should be changed to ‘Functionalization of silicone surface with drug and polymers for regulating capsule contracture’.

Response: Thank you for your comment. We have changed the title according to your comments.

Line 229-230, why the sentence is yellow highlighted?

Response: Thank you for your comments. We have checked the yellow highlighted which already corrected.

Line 313, ‘Polymers’ should be changed to ‘polymers’.

Response: We have changed ‘Polymers’ to ‘polymers’.

Line 62-64, different kinds of surface modification of silicones have been reported. Several studies (doi.org/10.1016/j.actbio.2011.02.001; doi.org/10.1021/la303438t; doi.org/10.3390/polym11020305) are recommended to be included regarding to this point.

Response: Thank you for the suggestion. We have cited the articles in the references at no. 20, 21, 22.

The resolution of figure 4, 6, and 13 should be improved.

Response: Thank you for the recommendation. We have improved the resolution of figure 4, 6, and 13.

There is no proper figure caption for figure 4-6.

Response: Thank you for the comment. Proper figure caption of figure 4-6 have been added.

Proper copyright statement should be added in some of the figure caption.

Response: Thank you for the suggestion. Copyright of the figures have been mentioned in figures captions.

Figure 9, some of the text is covered by the arrow.

Response: Thank you for the comment. We have perfectly placed the arrow to clear view the text for figure 9.

Section 4 is the essential section in this review since Polymers is polymer journal. The authors should improve this section to a higher level. For example, this section is divided into spider silk, poly(glucerol monoethacrylate), and methacryloxyethyl phosphorylcholine (MPC), which is not proper. I suggest it could be divided into three categories: natural polymer (proteins, polysaccharide, and so on), neutral hydrophilic polymer (PEG, poly(glucerol monoethacrylate, and so on), and zwitterionic polymer (poly(methacryloxyethyl phosphorylcholine), poly(carboxybetaine methacrylate), and so on).

Response: Thank you for the suggestion and recommendation. According to your suggestion, we have arranged the section 4 with three categories: natural polymer, neutral hydrophilic polymer and zwitterionic polymer. For improve the section in higher level we have also included three polymers namely IL-4 (protein) in section 4.1.2. line no 428-441, as natural polymers. PEG as neutral hydrophilic polymer in section 4.2.2 line no. 454-462; and Poly (carboxybetaine methacrylate) & poly(sulfobetaine methacrylate) as zwitterionic polymer in section 4.2.2 line no. 491-503.

Reviewer 2 Report

This review article is appropriate for the journal but needs some improvements:

  1. The application (breast cancer) should be specified in the title.

  1. Check the authors’ affiliations to correct some “strange” mistakes:

Depatment -> Department

Republic of Kore -> Korea

  1. Silicone is widely used in surgery, not only plastic surgery; an important example is given by scleral buckling in ophthalmology, please read and cite these papers in the Introduction:

The use of polymers in the treatment of retinal detachment: current trends and future perspectives. Polymers 2010;2:286-322

Scleral buckling biomaterials and implants for retinal detachment surgery. Medical Engineering and Physics 2010;32:945-956   

  1. The typical characteristics of “standard” silicone (e.g. Mw, density…) for breast surgery should be included in this review.

  1. A market perspective would be useful – what of the described options are currently available to surgeons? And what about the stage of regulatory approval?

  1. Language checking is recommended to correct some mistakes, for example:

- section 4.1: is “Poly (glucerol monoethacrylate)” correct? I guess “poly(glycerol monoethacrylate)”

Author Response

This review article is appropriate for the journal but needs some improvements:

Response: Thank you for your comments for the improvement of article.

The application (breast cancer) should be specified in the title.

Response: Thank you for the suggestion and comment. Since capsule contracture is most severe complication in breast surgery, our article highlights the modification of silicone implant using drugs and polymers to inhibit capsule formation. We have mentioned this specific application (capsule contracture) in the title.

Check the authors’ affiliations to correct some “strange” mistakes: Depatment -> Department. Republic of Kore -> Korea

Response: We have corrected the mistakes.

Silicone is widely used in surgery, not only plastic surgery; an important example is given by scleral buckling in ophthalmology, please read and cite these papers in the Introduction: The use of polymers in the treatment of retinal detachment: current trends and future perspectives. Polymers 2010;2:286-322. Scleral buckling biomaterials and implants for retinal detachment surgery. Medical Engineering and Physics 2010;32:945-956   

Response: Thank you for your suggestion. According to the suggestions, I have read the articles and cited in the introduction section. The reference no. 1 and 2 belongs the cited articles.

The typical characteristics of “standard” silicone (e.g. Mw, density…) for breast surgery should be included in this review.

Response: Thank you for the comment. According to your suggestion, we have included the typical characteristics of silicone for breast surgery in details in the introduction, section 1 line no. 75-84. Moreover, we have mentioned the breast implant characteristics based on FDA approval.

A market perspective would be useful – what of the described options are currently available to surgeons? And what about the stage of regulatory approval?

Response: Thank you for the suggestions. Based on your comments, we have discussed breast implant in respect of market perspective in the introduction, section 1 line no 47-74 in brief. Moreover, we have included in detail the available options for breast implant and their demand in market.

Language checking is recommended to correct some mistakes, for example:- section 4.1: is “Poly (glucerol monoethacrylate)” correct? I guess “poly(glycerol monoethacrylate)”

Response: Thank you for the comment. We have corrected the spelling errors. We have also checked the language from english editing service

Round 2

Reviewer 1 Report

The authors did serious revison based on my comments. However, several minor points (basically about the format) have to be solved before publication. I will suggest it for publication, but the authors should address it during the proof.

  1. Title: the font for 'drugs' is strange.
  2. Line 191, 'Surface Modification using Drugs' to 'Surface modification using drugs'. Line 407, 'Spider Silk' to 'Spider silk'. Please check all.
  3. Line 467, 'Methacryloxyethyl phosphorylcholine (MPC)' to 'poly(methacryloxyethyl phosphorylcholine) (pMPC)'.